# Mechanical Response of MEMS Suspended Inductors under Shock Using the Transfer Matrix Method

**DOI:** 10.3390/mi14061187

**Published:** 2023-06-01

**Authors:** Tianxiang Zheng, Lixin Xu

**Affiliations:** School of Mechatronical Engineering, Beijing Institute of Technology, Beijing 100081, China; lxxu@bit.edu.cn

**Keywords:** MEMS suspended inductors, shock load, transfer matrix method, dynamic response

## Abstract

MEMS suspended inductors are susceptible to deformation under external forces, which can lead to the degradation of their electrical properties. The mechanical response of the inductor to a shock load is usually solved by a numerical method, such as the finite element method (FEM). In this paper, the transfer matrix method of linear multibody system (MSTMM) is used to solve the problem. The natural frequencies and mode shapes of the system are obtained first, then the dynamic response by modal superposition. The time and position of the maximum displacement response and the maximum Von Mises stress are determined theoretically and independently of the shock. Furthermore, the effects of shock amplitude and frequency on the response are discussed. These MSTMM results agree well with those determined using the FEM. We achieved an accurate analysis of the mechanical behaviors of the MEMS inductor under shock load.

## 1. Introduction

The development of MEMS technologies introduces new approaches and designs to the fabrication of inductors in order to attain higher quality factors that are required for typical microwave circuits [1]. A MEMS inductor can be separated from the substrate by etching the sacrificial layer or the substrate, which is called the MEMS suspended inductor [2,3,4,5,6]. Although this structure decreases substrate loss in terms of electrical performance [1], it is sensitive to external forces, as the spiral is movable in terms of structural dynamics, just as MEMS capacitive accelerometers are [7]. The most severe external force that MEMS devices are subjected to during operations is mechanical shock. In particular, the normal shock acceleration during the drop process is around 500 g (g is the acceleration of Earth’s gravity); during the launch step of a high-speed vehicle, the devices are subjected to an acceleration that is much greater than 10,000 g [7,8]. This means that in some probable working situations, under such high-g shock loads, excessive deformation of the inductor might result in the degradation of the electrical properties, with some parts yielding or even fracturing, which is the most serious failure. There have been some research studies on MEMS suspended inductors in shock environments. Ribas et al. [2], Lin et al. [5] Hsieh et al. [6], and Dahlmann et al. [9], respectively, used the finite element program called ANSYS to examine the deformation of MEMS suspended inductors under shock acceleration, in addition to analyzing the electrical properties. In these studies, as the shock loads were below the drop level, the deformations of the inductors were almost negligible, and the researchers did not investigate further. The finite element method (FEM) mentioned above is generally used when the problems are too complicated to be solved by an analytical method, such as the deformation of three-axis MEMS gyroscopes [10].

On the other hand, researchers have proposed some analytical models of MEMS devices under shock [11]. Jiang et al. [12], Heish et al. [6], and Dahlmann et al. [9] compared the inductor to a cantilever beam and considered that the maximum displacement of the free end of the beam was equivalent to the maximum deformation of the inductor spiral. These treatments were rough and their results were incorrect. Likewise, Srikar et al. [13] proposed another equivalent model for treating a MEMS device as a single-degree-of-freedom (SDOF) system and obtained damage criteria formulations based on displacement and stress. Xu et al. [14] and Peng et al. [15] have each proposed more accurate formulations for describing shock acceleration based on the SDOF model. Wang et al. [16], Lian et al. [17], and Fathalilou et al. [18] have analyzed the shock resistance performance of MEMS gyroscopes and MEMS shock switches using a dual-vibrator model, an extension of the SDOF model, respectively. Fang et al. [19] and Singh et al. [20] combined both the SDOF model and the microbeam model to analyze the shock resistance of MEMS micromirrors and MEMS switches. Sundaram et al. [21] proposed a combined experimental–analytical approach by focusing on equating key parts of the device to the SDOF systems and they investigated the failure of a tunable diffraction grating under shock and vibration. Xu and Li [22,23,24] have carried out many meaningful studies. They equated the inductor to an SDOF system to obtain an equivalent acceleration, considered the inductor as a bar system, determined its deformation and stress by solving the super-stationary equations, and analyzed the mechanical reliability of the structure. The natural frequency of the structure, however, was established by the FEM; therefore, it was not a fully analytical solution. Moreover, they did not discuss the relationship between the shock and the response in detail, but only qualitatively. In fact, most analytical studies on microstructure mechanics have focused on a class of devices represented by microbeams [13,25,26,27,28], with little study on MEMS suspended inductors. Therefore, the situation calls for an accurate analytical method to analyze the response of the MEMS inductor under shock and to determine whether the inductor has failed since understanding the dynamics of a device is critical to determine its failure.

A MEMS suspended inductor could be regarded as a multibody system composed of rigid and flexible bodies in structural mechanics. In the research on multibody system dynamics, Rui et al. [29,30,31] systematically proposed the transfer matrix method for linear multibody systems (MSTMM) after more than 20 years of continuous refinement. This method introduces a new approach to analyzing the dynamics of a multibody system by establishing transfer relationships between the state vectors of each component. In addition, the authors applied the method to the dynamic analysis of laser gyros, piezoelectric actuators, rocket boosters, etc [32,33,34,35,36,37].

This paper describes in detail the use of the MSTMM to investigate the dynamic response of the MEMS suspended inductor under high-g shock. The natural frequencies and mode shapes were obtained according to the geometry and material parameters, and the dynamic response of the structure was solved by modal superposition to obtain the deformation and stress, as well as the maximum shock acceleration when the yield strength was reached.

## 2. Modeling and Method

### 2.1. Simplification of the Structure of the Inductor

Before the mechanical analysis, the real structure should be simplified, based on the characteristics of the individual problem, in order to skip over the unnecessary details and retain the basic features. The following describes the assumptions and simplifications used in building the mechanical model of the MEMS suspended planar inductor. A typical MEMS suspended planar inductor is made up of three parts: a spiral, a substrate, and two pillars, as shown in Figure 1. The spiral could be square-polygonal or circular in shape. In this paper, we investigate a 1.5-turn square spiral made of copper with a silicon substrate. The spiral was supported by two pillars on the substrate, which also connected the signal transmission lines. MEMS inductors, which are designed in the same layout but manufactured using different processes, might not be identical in several aspects, including metal thickness, surface flatness, silicon resistivity, silicon dioxide thickness, residual stress [38], and so on. Here, it is assumed that the three-dimensional dimensions of the metal spiral are the correct sizes in the design layout, in the sense that the spiral strips are cuboids in geometry.

Considering the shock, the load causes a sudden acceleration that acts on the object, which can be described by an acceleration–time curve and has three basic characteristics, e.g., strength, shape, and frequency. Shock loads in different environments are generally highly varied, even when generated by impact equipment used in the laboratory. The shock acceleration can usually be approximated by a half-sine pulse of the shock peak, which allows for the uniform analysis of a large class of shock problems, as shown in Figure 2. For MEMS devices, regardless of how the shock load is generated, it can be thought of as a distributed acceleration pulse acting on the package and substrate, characterized by a short duration (the package is assumed to have minimal influence on the strength and shape of the shock load [13]). Furthermore, because the damping of the system does not absorb enough energy in the duration, the damping effect could be ignored.

In the following, the MSTMM is used to analyze the problem of the dynamic response of the MEMS inductor under the shock load. The analysis could be organized into several steps. First, the system is decomposed into six components; next, the transfer equation of each component can be established. Each strip satisfies the definition of a bar so that it can be reduced to its axis. The connected parts of the adjoining strips are represented by the intersection of the axes, and the action points of the loads are transferred to the axes, as illustrated in Figure 3. The support pillars that connect the spiral to the substrate cannot be equated to its axis. In terms of this problem, it is considered that the pillars, together with the substrate, impose fixed constraints on the spiral due to their minimal deformation under shock loads, thus allowing them to be treated as rigid bodies.

It is clear that the spiral is the movable part of the structure and the direction perpendicular to the plane (in which the spiral is located) is the most prominently affected by the shock load. When subjected to the load in the normal direction, each bar undergoes deformation, which is mainly a combination of bending and torsion. Even though stress and deformation are more complicated, as long as the deformation is small and the bar remains linearly elastic, it can be assumed that the various basic deformation forms are independent of each other, which is the principle of the superposition of the combined deformation of the bar.

### 2.2. Free Vibration Characteristics of the Inductor

The next step is to determine the vibration characteristics of the spiral, including free and forced vibration characteristics. The free vibration characteristic is analyzed first; this includes the natural frequencies and mode shapes. Consequently, the transfer matrix of each component needs to be obtained by the MSTMM. Moreover, considering that the six bars have the same physical and material parameters, except the length, only the transfer matrix of the first bar needs to be derived, and the others can be acquired in the same manner.

The transfer equation of the bar is derived from the motion-governing equations of bending and torsion. Equation (Equation 1) is the partial differential equation, where the bar is bent, which describes the lateral free vibration of the bar, and is called the Euler–Bernoulli beam equation. In addition, according to the MSTMM, each element has its own stationary inertial Cartesian coordinate system. The origin is located at the endpoint that is first encountered along the transfer direction [31].
(1)$$EI\frac{\partial^{4}z}{\partial x^{4}}+\bar{m}\frac{\partial^{2}z}{\partial t^{2}}=0$$
where $\bar{m}$ is the linear density of the bar and EI is the bending stiffness of the beam; it is more convenient to set
$$z\left(x,t\right)=Z\left(x\right)e^{i\omega t}$$

Substituting the above into Equation (Equation 1), in the modal coordinates, there is
(2)$$\frac{\partial^{4}Z\left(x\right)}{\partial x^{4}}-\frac{\bar{m}\omega^{2}}{EI}Z\left(x\right)=0$$

The general solution of Equation (Equation 2) is
(3)$$Z\left(x\right)=C_{1}\cos\beta x+C_{2}\sin\beta x+C_{3}\cosh\beta x+C_{4}\sinh\beta x$$
where $\beta =\sqrt[{4}]{\frac{\bar{m}\omega^{2}}{\left(EI\right)}}.$

Considering the knowledge of geometry and the mechanics of materials,
(4)$$\Theta_{y}=\frac{dZ(x)}{dx},\;M_{y}=EI\frac{d\Theta_{y}}{dx},\;Q_{z}=\frac{dM_{y}}{dx}$$
where $\Theta_{y}$ denotes the angle displacement and $Q_{z}$ and $M_{y}$ denote the interior force and torque in the modal coordinates.

Substituting Equation (Equation 3) into Equation (Equation 4), and arranging the outcome into a matrix,
(5)$${\left[\begin{array}{c} Z\\ \Theta_{y}\\ M_{y}\\ Q_{z} \end{array}\right]}_{x}=\left[\begin{array}{cccc} \cosh\beta x & \sinh\beta x & \cos\beta x & \sin\beta x\\ \beta \sinh\beta x & \beta \cosh\beta x & -\beta \sin\beta x & \beta \cos\beta x\\ EI\beta^{2}\cosh\beta x & EI\beta^{2}\sinh\beta x & -EI\beta^{2}\cos\beta x & -EI\beta^{2}\sin\beta x\\ EI\beta^{3}\sinh\beta x & EI\beta^{3}\cosh\beta x & EI\beta^{3}\sin\beta x & -EI\beta^{3}\cos\beta x \end{array}\right]\left[\begin{array}{c} C_{1}\\ C_{2}\\ C_{3}\\ C_{4} \end{array}\right]$$
the above can be abbreviated as
(6)Z(x)=B(x)C
where Z(x) denotes the state vector at *x* in the modal coordinates, C is the matrix of the integral constants. The state vector Z is a column vector representing the mechanical state of any point in a multibody system; the elements are the (angular) displacement, the internal force, and the internal moment at that point.

In order to formally eliminate the integral constant vector C, the operations can be done as follows. when x=0, $C=B^{-1}\left(0\right)Z\left(0\right)$, so $Z\left(x\right)=B\left(x\right)B^{-1}\left(0\right)Z\left(0\right)=U\left(x\right)Z\left(0\right)$. In particular, when x=l (*l* is the length of the continuum axis of a particular component), Z(l)=U(l)Z(0). More generally, it can be written as
(7)$$Z_{O}=UZ_{I}$$
where U is the transfer matrix of the bending bar, which does lateral free vibration. The subscripts *O* and *I* indicate the output and input points of the component, respectively.

For the bar bending problem, there is
(8)$${\left[\begin{array}{c} Z\\ \Theta_{y}\\ M_{y}\\ Q_{z} \end{array}\right]}_{O}=\left[\begin{array}{cccc} S(\beta l) & \frac{T(\beta l)}{\beta } & \frac{U(\beta l)}{EI\beta^{2}} & \frac{V(\beta l)}{EI\beta^{3}}\\ \beta V(\beta l) & S(\beta l) & \frac{T(\beta l)}{EI\beta } & \frac{U(\beta )}{EI\beta^{2}}\\ EI\beta^{2}U\left(\beta l\right) & EI\beta V(\beta l) & S(\beta l) & \frac{T(\beta l)}{\beta }\\ EI\beta^{3}T\left(\beta l\right) & EI\beta^{2}U\left(\beta \right) & \beta V(\beta l) & S(\beta l) \end{array}\right]{\left[\begin{array}{c} Z\\ \Theta_{y}\\ M_{y}\\ Q_{z} \end{array}\right]}_{I}$$
where
$$\begin{array}{c} S\left(\beta x\right)=\frac{\cosh\left(\beta x\right)+\cos\left(\beta x\right)}{2},\;V\left(\beta x\right)=\frac{\sinh\left(\beta x\right)-\sin\left(\beta x\right)}{2}\\ U\left(\beta x\right)=\frac{\cosh\left(\beta x\right)-\cos\left(\beta x\right)}{2},\;T\left(\beta x\right)=\frac{\sinh\left(\beta x\right)+\sin\left(\beta x\right)}{2} \end{array}$$

Equation (Equation 9) represents the partial differential equation that arises when a round bar is twisted.
(9)$$GJ_{p}\frac{\partial^{2}\Theta_{x}}{\partial x^{2}}-\rho J_{p}\frac{\partial^{2}\Theta_{x}}{\partial t^{2}}=0$$
where ρ is the volume density, *G* is the shear modulus, and $J_{p}$ is the polar moment of inertia. In this problem, some changes must be made to the parameters in Equation (Equation 9) because the cross-section of the bar is not round but square. According to the related research of elasticity, $J_{p}$ in the torsional stiffness $GJ_{p}$ is replaced by the equivalent polar moment of inertia $J_{s}$ of a rectangular cross-section bar. Moreover, $J_{p}$ in $\rho J_{p}$ is replaced by the polar moment of inertia $J_{o}$ of the rectangular cross-section.

Similarly, in the modal coordinates,
(10)$$\frac{\partial^{2}\Theta_{x}}{\partial x^{2}}+\frac{\rho J_{o}\omega^{2}}{GJ_{s}}\Theta_{x}=0$$

The general solution to the above is
(11)$$\Theta_{x}=D_{1}\sin\gamma x+D_{2}\cos\gamma x$$
where $\gamma =\omega \sqrt{\frac{\rho J_{o}}{GJ_{s}}}.$

Considering the mechanical relationship of materials,
(12)$$M_{x}=GJ_{s}\frac{d\Theta_{x}}{dx}$$

Substituting Equation (Equation 11) into Equation (Equation 12) and arranging the outcome into a matrix,
(13)$${\left[\begin{array}{c} \Theta_{x}\\ M_{x} \end{array}\right]}_{x}=\left[\begin{array}{cc} \sin\gamma x & \cos\gamma x\\ \gamma GJ_{s}\cos\gamma x & -\gamma GJ_{s}\sin\gamma x \end{array}\right]\left[\begin{array}{c} D_{1}\\ D_{2} \end{array}\right]$$

Eliminating the integral constants matrix and writing it in the form of Equation (Equation 7),
(14)$${\left[\begin{array}{c} \Theta_{x}\\ M_{x} \end{array}\right]}_{O}=\left[\begin{array}{cc} \cos\gamma l & \frac{1}{\gamma GJ_{s}}\sin\gamma l\\ -\gamma GJ_{s}\sin\gamma l & \cos\gamma l \end{array}\right]{\left[\begin{array}{c} \Theta_{x}\\ M_{x} \end{array}\right]}_{I}$$

By combining Equations (Equation 8) and (Equation 14), we can derive the transfer matrix equation for a bending and twisting bar as follows:(15)$$Z_{O}=\left[\begin{array}{cc} U_{11} & U_{12}\\ U_{21} & U_{22} \end{array}\right]Z_{I}$$
where
$$\begin{array}{cc} Z_{O} & ={\left[\begin{array}{c} Z,\Theta_{x},\Theta_{y},M_{x},M_{y},Q_{z} \end{array}\right]}_{O}^{T}\\ Z_{I} & ={\left[\begin{array}{c} Z,\Theta_{x},\Theta_{y},M_{x},M_{y},Q_{z} \end{array}\right]}_{I}^{T} \end{array}$$
$$U_{11}=\left[\begin{array}{ccc} S(\beta l) & 0 & \frac{T(\beta l)}{\beta }\\ 0 & \cos\gamma l & 0\\ \beta V(\beta l) & 0 & S(\beta l) \end{array}\right],\;U_{12}=\left[\begin{array}{ccc} 0 & \frac{U(\beta l)}{EI\beta^{2}} & \frac{V(\beta l)}{EI\beta^{3}}\\ \frac{1}{\gamma GJ_{s}}\sin\gamma l & 0 & 0\\ 0 & \frac{T(\beta l)}{EI\beta } & \frac{U(\beta l)}{EI\beta^{2}} \end{array}\right]$$
$$U_{21}=\left[\begin{array}{ccc} 0 & -\gamma GJ_{s}\sin\gamma l & 0\\ EI\beta^{2}U\left(\beta l\right) & 0 & EI\beta V(\beta l)\\ EI\beta^{3}T\left(\beta l\right) & 0 & EI\beta^{2}U\left(\beta l\right) \end{array}\right],\;U_{22}=\left[\begin{array}{ccc} \cos\beta l & 0 & 0\\ 0 & S(\beta l) & \frac{T(\beta l)}{\beta }\\ 0 & \beta V(\beta l) & S(\beta l) \end{array}\right]$$

In fact, the transfer equations of all types of elements can be written in this form. For convenience, the endpoints of all components are marked counterclockwise from 0 to 6, and the corresponding state vectors are marked $Z_{0}$ to $Z_{6}$, so that
(16)$$Z_{j}=U_{j}Z_{j-1}\;\left(j=1,2,\cdots ,6\right)$$
where $U_{j}$ denotes the transfer matrix of the *j*-th component and $Z_{j}$ denotes the state vector of the *j*-th point.

Note that the output point of one component is exactly the input point of the next component, which means that the state vectors of these two points should be equal, taking into account the rotation of the coordinate system. For a chain system, such as the inductor spiral, the total system transfer equation can be obtained by assembling all transfer equations in a counter-clockwise order. Thus, the total transfer equation of the spiral is
(17)$$\begin{array}{cc} Z_{6} & =U_{6}\cdots HU_{j}\cdots HU_{1}Z_{0}\\ \; & =U_{all}Z_{0} \end{array}$$
where $U_{all}$ is the total transfer matrix of the system and is a 6 by 6 matrix, H represents the direction cosine matrix that is rotated 90 degrees counterclockwise. It can be seen that the total transfer equation of the system only involves the state vectors of the boundary points and not the state vectors of the connection points. Moreover, the elements in the total transfer matrix $U_{all}$ are simply functions of the structure parameters and the natural frequency. By substituting the boundary conditions into Equation (Equation 17), the homogeneous linear equations, which are called characteristic equations, can be extracted. In general, half of the elements in the boundary point state vector are unknown and half are known. In the study, when both ends are fixed,
(18)$$Z_{0}={\left[0,0,0,M_{x},M_{y},Q_{z}\right]}_{0}^{T},\;Z_{6}={\left[0,0,0,M_{x},M_{y},Q_{z}\right]}_{6}^{T}$$

Substituting the above into Equation (Equation 17), the corresponding elements in $Z_{0},Z_{6},U_{all}$ form the characteristic equation.
(19)$${\left[\begin{array}{c} 0\\ 0\\ 0 \end{array}\right]}_{6}=\left[\begin{array}{ccc} U_{14} & U_{15} & U_{16}\\ U_{24} & U_{25} & U_{26}\\ U_{34} & U_{36} & U_{36} \end{array}\right]{\left[\begin{array}{c} M_{x}\\ M_{y}\\ Q_{z} \end{array}\right]}_{0}$$

The force $Q_{z}$ and the torque $M_{x},M_{y}$ must exist, So the determinant of the coefficient matrix $U^{\prime }$ should be zero, $\det U^{\prime }=0$. The natural frequency $\omega_{k}\left(k=1,2,\cdots \right)$ can be obtained by solving the determinant in which the only unknown quantity is the natural frequency of the system. In the case that $Q_{z,0}=1$, the values of $M_{x,0},M_{y,0}$ can be derived by solving non-homogeneous linear equations, which are part of the elements of Equation (Equation 19). In this way, the state vector of the start point, $Z_{0}$, is obtained, and the values of $Z_{1}$ to $Z_{6}$ can also be obtained by substituting $Z_{0}$ with Equation (Equation 16). The state vector at any position of each element is given by $Z_{j}\left(x\right)=U_{j}\left(x\right)Z_{j-1}$. Moreover, the displacements and angular displacements in all of the $Z_{j}\left(x\right)$ constitute the modal shapes of the system.

### 2.3. Dynamics Equation of the Inductor

The following describes how to solve the response of the spiral under shock acceleration. The dynamic equations of each component can be written in the following form [31]:(20)$$M_{j}v_{j,tt}+K_{j}v_{j}=f_{j}\;\left(j=1,\cdots ,6\right)$$
where $M_{j}$ and $K_{j}$ are the parameter matrices, which can be called the mass matrix and stiffness matrix, following the habits of vibration mechanics. $v_{j}$ is the column matrix of displacement (angular displacement). $v_{j,tt}$ and $v_{j,t}$ denote the two-order derivative and one-order derivative of $v_{j}$, with respect to time *t*. $f_{j}$ is the column matrix of the force (torque) acting on the *j*-th body (*j* is the body index in the multibody system). Likewise, air damping is not taken into account.

The body dynamics equation of the bending and twisting bar can be obtained by arranging the partial differential Equations (Equation 1) and (Equation 9), taking into account the force to the matrix.
(21)$$\left[\begin{array}{cc} {\bar{m}}_{j} & \\ \; & {\left(\rho J_{o}\right)}_{j} \end{array}\right]{\left[\begin{array}{c} z\\ \theta_{x} \end{array}\right]}_{tt}+\left[\begin{array}{cc} {\left(EI_{y}\right)}_{j}\frac{\partial^{4}}{\partial x^{4}} & \\ \; & -{\left(GJ_{p}\right)}_{j}\frac{\partial^{2}}{\partial x^{2}} \end{array}\right]\left[\begin{array}{c} z\\ \theta_{x} \end{array}\right]=\left[\begin{array}{c} {\bar{m}}_{j}a\\ 0 \end{array}\right]$$
where *a* is the shock acceleration.

The total body dynamic equation of the system can be easily obtained by arranging the body equations of all components in turn,
(22)$$Mv_{tt}+Kv=f$$
where
$$\begin{array}{cc} \; & M=\mathrm{diag}\left(M_{1},M_{2},\cdots ,M_{j}\right),\;K=\mathrm{diag}\left(K_{1},K_{2},\cdots ,K_{j}\right)\\ \; & v={\left[v_{1}^{T},v_{2}^{T},\cdots ,v_{j}^{T}\right]}^{T},\;f={\left[f_{1},f_{2},\cdots ,f_{j}\right]}^{T} \end{array}$$

### 2.4. Orthogonality of Eigenvectors

In the case of free vibration, the elements in the column matrix of displacement v can be represented in modal coordinates, $v=Ve^{i\omega t}$. Then V, v in the modal coordinates can be obtained, as shown in the following formula, which is denoted as the eigenvector of the system.
(23)$$V^{k}={\left[V_{1}^{k}{}^{T},V_{2}^{k}{}^{T},\cdots ,V_{j}^{k}{}^{T}\right]}^{T}$$
where *k* is the sequence number of the natural frequency of the structure.

According to the characteristics of the parameter matrices M and K, it can be proved that the two pairs of inner products are equal, respectively, which is called the symmetry of eigenvectors, with respect to M and K [31].
(24)$$\langle MV^{k_{1}},V^{k_{2}}\rangle =\langle V^{k_{1}},MV^{k_{2}}\rangle ,\;\langle KV^{k_{1}},V^{k_{2}}\rangle =\langle V^{k_{1}},KV^{k_{2}}\rangle$$

The sufficient mathematical proof of the symmetry of the eigenvectors is given in Appendix A.

For the free vibration of the *k*-th mode, $v=V^{k}e^{i\omega_{k}t}.$ Substituting v into Equation (Equation 22), in the case of free vibration,
(25)$$\omega_{k}^{2}MV^{k}=KV^{k}$$
so
(26)$$\begin{array}{cc} \omega_{k_{1}}^{2}\langle MV^{k_{1}},V^{k_{2}}\rangle & =\langle \omega_{k_{1}}^{2}MV^{k_{1}},V^{k_{2}}\rangle =\langle KV^{k_{1}},V^{k_{2}}\rangle \\ \; & =\langle V^{k_{1}},KV^{k_{2}}\rangle =\langle V^{k_{1}},\omega_{k_{2}}^{2}MV^{k_{2}}\rangle \\ \; & =\omega_{k_{2}}^{2}\langle MV^{k_{1}},V^{k_{2}}\rangle \end{array}$$

Eventually, there is
(27)$$\left(\omega_{k_{1}}^{2}-\omega_{k_{2}}^{2}\right)\langle MV^{k_{1}},V^{k_{2}}\rangle =0$$
when $\omega_{k_{1}}\neq \omega_{k_{2}}$, $\langle MV^{k_{1}},V^{k_{2}}\rangle =0$. The same procedure can be easily adapted to prove that $\langle KV^{k_{1}},V^{k_{2}}\rangle =0\left(\omega_{k_{1}}\neq \omega_{k_{2}}\right)$. These show that the mode shapes with respect to different frequencies are orthogonal to each other for the mass and stiffness matrices. For convenience, they can be written as
(28)$$\langle MV^{k_{1}},V^{k_{2}}\rangle =\delta_{k}M_{k},\;\langle KV^{k_{1}},V^{k_{2}}\rangle =\delta_{k}K_{k}$$
where
$$\delta_{k}=\left\{\begin{array}{cc} 0 & k_{1}\neq k_{2}\\ 1 & k_{1}=k_{2} \end{array}\right.$$

### 2.5. The Dynamic Response of the Inductor

Having established the orthogonality of the eigenvectors V, the displacement response v can be expressed by the product of the generalized coordinate q(t) and the eigenvectors V, as seen below, according to the modal superposition method [31].
(29)$$v=\sum_{k=1}^{n}V^{k}q^{k}\left(t\right)$$

Substituting Equation (Equation 29) into the total body dynamics Equation (Equation 22), then
(30)$$\sum_{k=1}^{n}MV^{k}{\ddot{q}}^{k}\left(t\right)+\sum_{k=1}^{n}KV^{k}q^{k}\left(t\right)=f$$

Taking the inner product on both sides with the eigenvector $V^{k}$ and using its orthogonality,
(31)$${\ddot{q}}^{k}\left(t\right)+\omega_{k}^{2}q^{k}\left(t\right)=\frac{\langle f,V^{k}\rangle }{M_{k}}\;\left(k=1,2,\cdots ,n\right)$$
let
(32)$$f_{k}\left(t\right)=\langle f,V^{k}\rangle$$
which is a single-valued function that only depends on time *t*.

Using Duhamel’s integral to obtain the solution of Equation (Equation 31), on the condition that the initial position and initial velocity of the system in this problem are both zero, it holds that
(33)$$q^{k}\left(t\right)=\frac{1}{M_{k}\omega_{k}}\int_{0}^{t}f_{k}\left(\tau \right)\sin\omega_{k}\left(t-\tau \right)d\tau$$

Finally, the dynamic response of the system is
(34)$$v=\sum_{k=1}^{n}V^{k}\frac{1}{M_{k}\omega_{k}}\int_{0}^{t}f_{k}\left(\tau \right)\sin\omega_{k}\left(t-\tau \right)d\tau$$

### 2.6. The Maximum Z-Axis and Angular Displacements

After obtaining the dynamic response of the inductor, the maximum displacement problem and von Mises stress are discussed. Taking out the Z-axis displacement function $z_{j}\left(x\right)$ of each element from the response matrix v, there is
(35)$$z_{j}\left(x,t\right)=\sum_{k=1}^{n}Z_{j}^{k}\left(x\right)\frac{1}{M_{k}\omega_{k}}\int_{0}^{t}f_{k}\left(\tau \right)\sin\omega_{k}\left(t-\tau \right)d\tau$$

To calculate $f_{k}\left(t\right)$ in the above equation, we substitute the function expression of the half-sine wave shock pulse, $a\left(t\right)=a_{0}\sin\omega_{f}t\left(t\leq \tau \right)$, into Equation (Equation 32),
(36)$$f_{k}\left(t\right)=a_{0}\sin\omega_{f}t\sum_{j=1}^{6}\int_{0}^{l_{j}}{\bar{m}}_{j}Z_{j}^{k}\left(x\right)dx=N_{k}a_{0}\sin\omega_{f}t$$
where $N_{k}$ depends on $\omega_{k}$.

Substituting the above into Equation (Equation 35), there is
(37)$$z_{j}\left(x,t\right)=a_{0}\sum_{k=1}^{n}Z_{j}^{k}\left(x\right)\frac{N_{k}}{M_{k}\omega_{k}}\int_{0}^{t}\sin\omega_{f}\tau \sin\omega_{k}\left(t-\tau \right)d\tau =\sum_{k=1}^{n}z_{j}^{k}\left(x,t\right)$$

It can be seen that, for any point on the spiral, its response is similar to the forced vibration of an undamped single-degree-of-freedom system. Calculating the integral, each component in Equation (Equation 37) can be expressed as
(38)$$z_{j}^{k}\left(x,t\right)=a_{0}\phi_{k}\left(t\right)Z_{j}^{k}\left(x\right)\frac{N_{k}}{M_{k}\omega_{k}^{2}}$$
where
(39)$$\phi_{k}\left(t\right)=\left\{\begin{array}{cc} \frac{1}{1-{\left(\frac{\omega_{f}}{\omega_{k}}\right)}^{2}}\left(\sin\omega_{f}t-\frac{\omega_{f}}{\omega_{k}}\sin\omega_{k}t\right) & t\leq \tau \\ -\frac{2}{1-{\left(\frac{\omega_{f}}{\omega_{k}}\right)}^{2}}\frac{\omega_{f}}{\omega_{k}}\cos\frac{\omega_{k}\tau }{2}\sin\omega_{k}\left(t-\frac{\tau }{2}\right) & t>\tau \end{array}\right.$$

Likewise, the angular displacement $\theta_{x,j}$ of each element has a similar mathematical form.
(40)$$\theta_{x,j}=a_{0}\sum_{k=1}^{n}\Theta_{x,j}^{k}\left(x\right)\frac{N_{k}}{M_{k}\omega_{k}}\int_{0}^{t}\sin\omega_{f}\tau \sin\omega_{k}\left(t-\tau \right)d\tau =\sum_{k=1}^{n}\theta_{x,j}^{k}\left(x,t\right)$$
where
(41)$$\theta_{x,j}^{k}\left(x,t\right)=a_{0}\phi_{k}\left(t\right)\Theta_{x,j}^{k}\left(x\right)\frac{N_{k}}{M_{k}\omega_{k}^{2}}$$

It is generally accepted that the maximum response caused by the shock load is more meaningful than other responses throughout the entire reaction process. It is the right time to discuss where and when the maximum displacement and stress occur. The displacement response is the sum of the contribution of each mode shape. In most cases, the lowest frequency portion makes the largest contribution, and the higher frequency portions tend to decrease significantly [39]. Moreover, as can be seen from Equations (Equation 38) and (Equation 41), the roles, in terms of the functions of position and time in the contribution of the modal shapes, are independent. Therefore, it is only necessary to find the position and time corresponding to the maximum of the first-order portion. This specific position can be obtained by plotting the position function, and the specific moment needs to be discussed on a case-by-case basis. First, we discuss the case of t≤τ in Equation (Equation 39), take the derivative of the function, and set it equal to zero; we can obtain $\cos\omega_{f}t=\cos\omega_{k}t$. Considering that $\omega_{f}t\leq \pi$, $\omega_{f}t$ should meet $\omega_{f}t=\frac{2n\pi }{\omega_{f}+\omega_{k}}$ where $4n-3<\frac{\omega_{k}}{\omega_{f}}\leq 4n+1$ (n is a positive integer); in the second case, the problem is simpler, where t should meet $t=\frac{\pi }{2}\left(\frac{1}{\omega_{f}}+\frac{1}{\omega_{k}}\right)$.

### 2.7. The Maximum von Mises Stress

With the column matrix displacement v obtained, the moment and stress can be calculated according to the relationship between displacement, moment, and stress. For each bar component, the bending moment on the cross-section at position *x* is
(42)$$m_{y}\left(x\right)=EI\frac{d^{2}z\left(x\right)}{dx^{2}}$$

The maximum tensile stress appears at the long side of the cross-section.
(43)$$\sigma_{max}\left(x\right)=\frac{m_{y}\left(x\right)z_{max}}{I_{y}}$$

Moreover, the torque on the cross-section at position *x* is
(44)$$m_{x}\left(x\right)=GJ_{s}\frac{d\theta_{x}}{dx}$$

The maximum shear stress appears at the midpoint of the long side of the cross-section.
(45)$$\tau_{max}\left(x\right)=\frac{m_{x}\left(x\right)}{\alpha bh^{2}}\;\left(\alpha =0.246\right)$$

Furthermore, the maximum von Mises stress appears at the midpoint of the long side of the cross-section.
(46)$$\sigma_{\;\mathrm{von}\;\mathrm{Mises}\;}\left(x\right)=\sqrt{\sigma_{\max}{\left(x\right)}^{2}+3\tau_{\max}{\left(x\right)}^{2}}$$

Substituting Equation (Equation 37) with (Equation 41) into the above equation, and making the necessary arrangements to obtain
(47)$$\sigma_{\;\mathrm{von}\;\mathrm{Mises}\;}\left(x,t\right)=a_{0}\sqrt{{\left(\frac{hE}{2}\sum_{k=1}^{n}\phi_{k}\left(t\right)\frac{N_{k}}{M_{k}\omega_{k}^{2}}\frac{d^{2}Z_{j}^{k}\left(x\right)}{dx^{2}}\right)}^{2}+3{\left(\frac{GJ_{s}}{\alpha bh^{2}}\sum_{k=1}^{n}\phi_{k}\left(t\right)\frac{N_{k}}{M_{k}\omega_{k}^{2}}\frac{d\Theta_{x,j}^{k}\left(x\right)}{dx}\right)}^{2}}$$
where *b* and *h* are the lengths of the long and short sides of the beam cross-section, respectively.

The position and time of the maximum von Mises stress are determined by the position and time of the maximum contribution of the first-order modal shape. So, only considering the case of *n* = 1, Equation (Equation 47) becomes
(48)$$\sigma_{\;\mathrm{von}\;\mathrm{Mises}\;}\left(x,t\right)\approx a_{0}\frac{N_{1}}{M_{1}\omega_{1}^{2}}\left|\phi_{1}\left(t\right)\right|\sqrt{{\left(\frac{hE}{2}\frac{d^{2}Z_{j}^{1}\left(x\right)}{dx^{2}}\right)}^{2}+3{\left(\frac{GJ_{s}}{\alpha bh^{2}}\frac{d\Theta_{x,j}^{1}\left(x\right)}{dx}\right)}^{2}}$$

This formula demonstrates that the maximum von Mises stress occurs at the same time as the displacement, and the position is determined by graphing the function.

## 3. Results and Discussion

This section mainly includes the results obtained by the MSTMM and the FEM to deal with the response of the MEMS suspended inductor under shock. The natural frequencies of the inductor were calculated, as well as the displacements and von Mises stresses under different loads. The accuracies of these data were verified by FEM. The material parameters used are listed below, including the parameters involved in the previous section. The Young’s modulus of copper is 128 GPa, the Poisson’s ratio is 0.34, and the density is 8900 kg/$m^{3}$. The cross-section of the strips had a width(b) of 20 μm and a height(h) of 10 μm. The spiral was 1.5 turns with a maximum outer diameter of 200 μm and a gap of 10 μm.

### 3.1. Natural Frequencies of the Inductor

The natural frequencies obtained by FEM pertained to all degrees of freedom in the system. For this problem, only the frequencies corresponding to the z-direction mode shapes were required. Therefore, these frequencies were selected and compared with the results obtained by MSTMM. As can be seen from Table 1 below, the natural frequencies of the MEMS inductor were relatively high, partly because the mass was generally very small. The errors between the data calculated by the two methods were small and should be due to the simplifications of the model.

### 3.2. Responses of the Inductor under Shock

A half-sine pulse load was selected with an amplitude of 10,000 g, a frequency of 10 kHz, and a direction along the negative z-axis, which meant that the shock load lasted 50 μs. Figure 4a,b shows, respectively, the displacement and the maximum Von Mises stress of each cross-section of the strips at a given moment (actually, these moments were the times at which the maximum values occurred). The responses of the displacements obtained by both methods were in good agreement. It is apparent that the maximum response of vertical displacement appeared at the intersection of components B3 and B4, namely P3. Moreover, the displacement results for B3 were larger than the FEM results (maximum relative error is 7.5%), which should be due to the approximation of the torsion equation.

For the von Mises stress, however, the situation became a little more complicated. Excluding all intersections and the surrounding region, the stresses exhibited relatively good agreement. However, at the intersections, the results obtained by FEM became particularly small. It indicated that stress concentrations appeared at these corners. The variation in the cross-section caused the stresses to be concentrated in the regions near the corners, which meant that the maximum values of the stresses were not at the midpoints of the long sides of the cross-sections. The previous method of calculating the stress was no longer valid. To solve this problem, a method was proposed to estimate the concentrated stress by vector-summing the shear and normal stresses of the two cross-sections of the two adjoining components corresponding to the corner point, and then calculating the von Mises stress; the results are shown in Table 2. It can be seen that the relative errors were small at P1, P4, and P5, where the stress concentration effects were more obvious; the relative errors were very large at P2 and P3, where the effects were not so obvious. Moreover, the results obtained by FEM were related to the mesh density: the denser the mesh, the higher the stress values. Thus, the maximum von Mises stress appeared at the intersection of elements B4 and B5, namely P5. The stresses at the first fixing points were very high as well.

Next, we present the results relating to the time at which the maximum value occurred. Here, an additional shock with a frequency of 70 kHz was chosen to verify the correctness of the MSTMM and to determine the timings of the maximum responses at different frequencies.

Figure 5 and Figure 6 present the time histories of the vertical displacements at position P3 and the von Mises stress at corner P5 on the spiral when subjected to shocks with frequencies of 10 kHz and 70 kHz, and they indicate that the shapes of the time histories of the responses were almost identical at all points if the response at each point was taken as the absolute value and normalized, which was exactly the function curve of |ϕ(t)| given in Equation (Equation 39). This also meant that the maximum displacement occurred almost at the same time as the maximum stress.

Looking first at Figure 5, the maximum response occurred during the forced vibration phase when the frequency of the shock was lower than the first-order natural frequency. The time of the maximum response obtained by the MSTMM coincided with the first peak of the curve, whereas the time obtained by the FEM corresponded to the second peak. This was due to the slight difference in the natural frequencies obtained by the two methods, which in turn affected the *n* in the expression of the times of the maximum being 1 and 2, respectively. The relative error of the maximum displacement response was 7.6% and the relative error of the stress was 4.0%. Looking at Figure 6, the maximum response occurred in the free vibration phase due to the ratio of the force and the first-order natural frequency being greater than 1. The relative error of the maximum displacement response was 5.1% and the relative error of the stress was 5.0%. In this subsection, we determine the time and position at which the maximum response of the inductor under shock load occurs, which is very important for the following analysis.

### 3.3. Effect of Shock Amplitude on Responses

Figure 7a,b show the maximum displacement and the maximum von Mises stress of the inductor under the half-sine shocks with the same frequency of 10 kHz but different amplitudes. It can be seen from these two figures that the values of the responses were indeed proportional to the amplitude of the shocks, as mentioned in the ’method’ section.

### 3.4. Effect of Shock Frequency on Responses

Figure 8 shows the maximum displacement and maximum von Mises stress of the inductor under half-sine shocks with the same amplitude of 10,000 g but different frequencies. Similarly, it can be seen that the displacement and the stress curves exhibit nearly identical shapes when normalized. The maximum displacement and stress both occurred at a frequency of 32 kHz. This means that the most dangerous situation would appear when the frequency of the force is about 32 kHz, which corresponds to a duration of 15.6 μs; in this case, an amplitude of 36,000 g would cause the von Mises stress to exceed the yield strength of the body copper by 200 MPa. It is also important to note that the displacement must be less than the suspended height. One more pillar could be added at P3 to prevent the spiral from fracturing. The advantage of the analytical method was shown when dealing with problems of different external forces, where only a few parameters related to the external forces needed to be changed. In contrast, the FEM had to repeat the entire process. For the cases in this subsection, the MSTMM took only 5 min, while the FEM took over 30 min.

From the previous discussion, it can be confirmed that it is possible to calculate the maximum displacement and maximum von Mises stress (within the yield strength) of the MEMS inductor under a shock of any amplitude and frequency. The responses at different frequencies with an amplitude of 10,000 g can be found in Figure 8a,b, respectively, and the values can be obtained by multiplying the corresponding ratios of the amplitudes.

## 4. Conclusions

In this paper, the response of the MEMS suspended inductor under high-g shock was investigated using the transfer matrix method of the linear multibody system, MSTMM. The free vibration characteristics of the system and the dynamic response under shock load were obtained. It was theoretically demonstrated that the positions where the maximum displacement and von Mises stress occurred can be determined independently of the shock load. Moreover, the procedure used to find the time of the maximum was explained; the method of identifying the maximum von Mises stress in the case of stress concentration was also discussed. The effect of the shock amplitude on the response was linear, and the response curves of the system were given for different shock frequencies. Therefore, the maximum response value to the shock load at a certain frequency at any amplitude could be quickly determined by looking up values in the frequency-response table and multiplying them by the amplitude ratios. This approach helps identify where the inductor may yield or fracture. Our work could serve as the foundation for designing shock-resistant MEMS suspended inductors and as a mechanical basis for further research on the relationship between changes in inductor electrical properties and shock loads.

## Figures and Tables

**Figure 1 micromachines-14-01187-f001:**
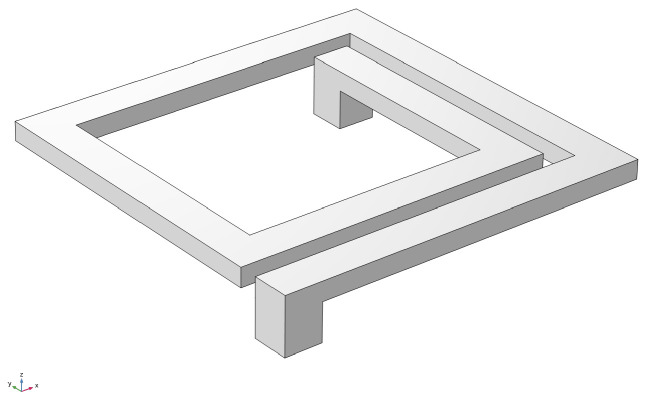
Schematic diagram of a MEMS planar spiral inductor except the subsubstrate.

**Figure 2 micromachines-14-01187-f002:**
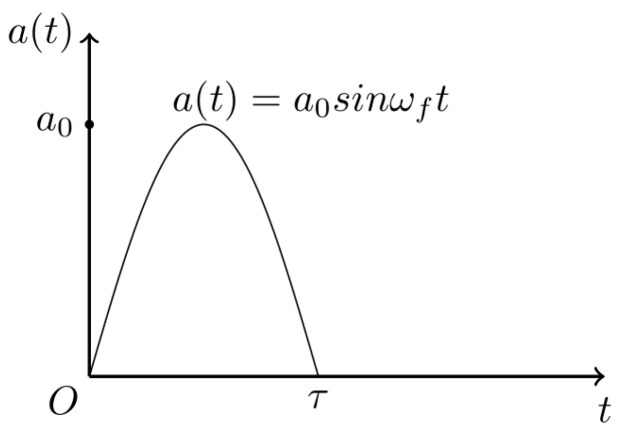
A half-sine acceleration pulse.

**Figure 3 micromachines-14-01187-f003:**
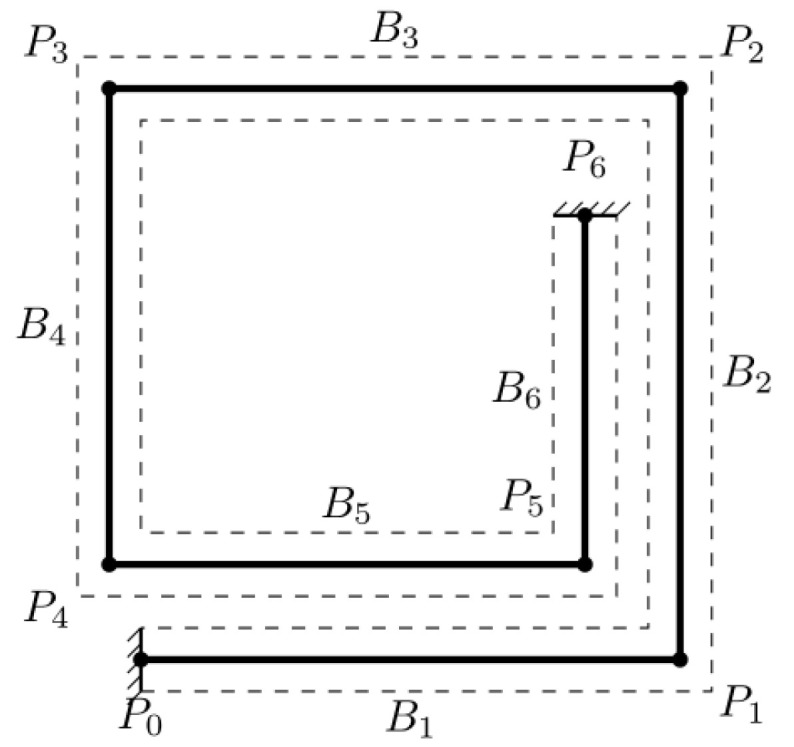
Simplified mechanics model of the inductor ($B_{i}$ denotes the *i*-th strip, $P_{i}$ denotes the *i*-th intersection).

**Figure 4 micromachines-14-01187-f004:**
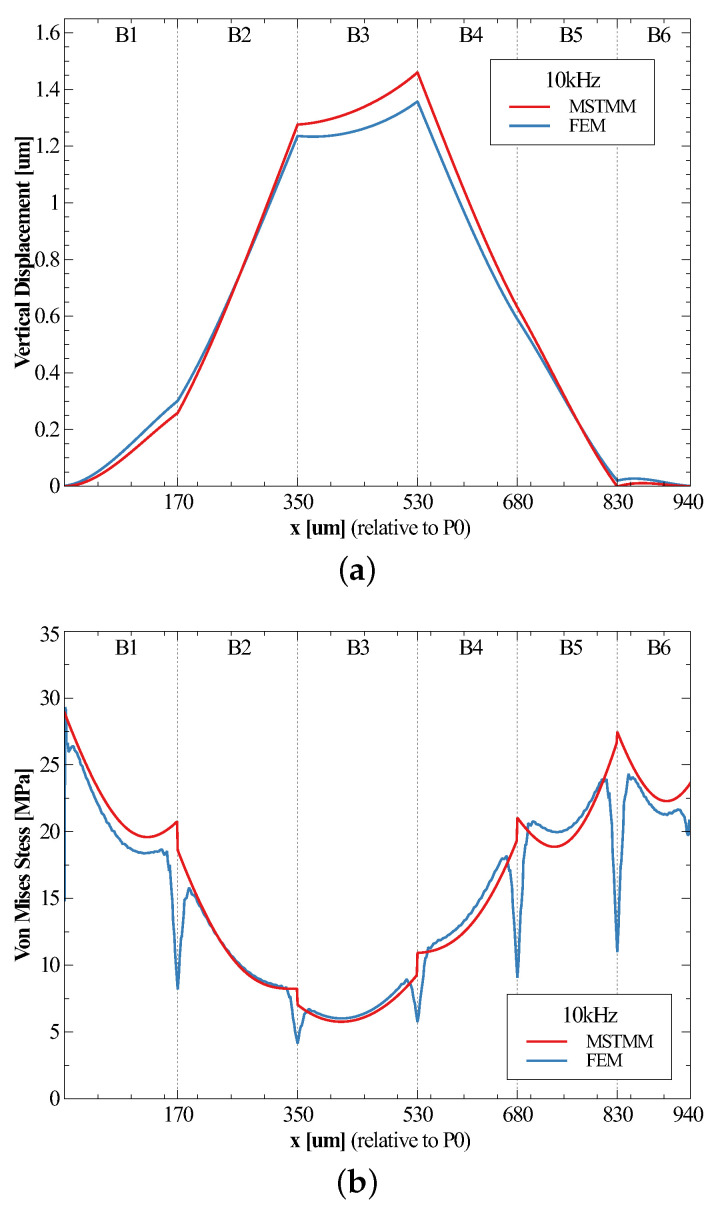
Vertical displacements (**a**) and the maximum von Mises stresses (**b**) in all cross-sections of the spiral at given moments.

**Figure 5 micromachines-14-01187-f005:**
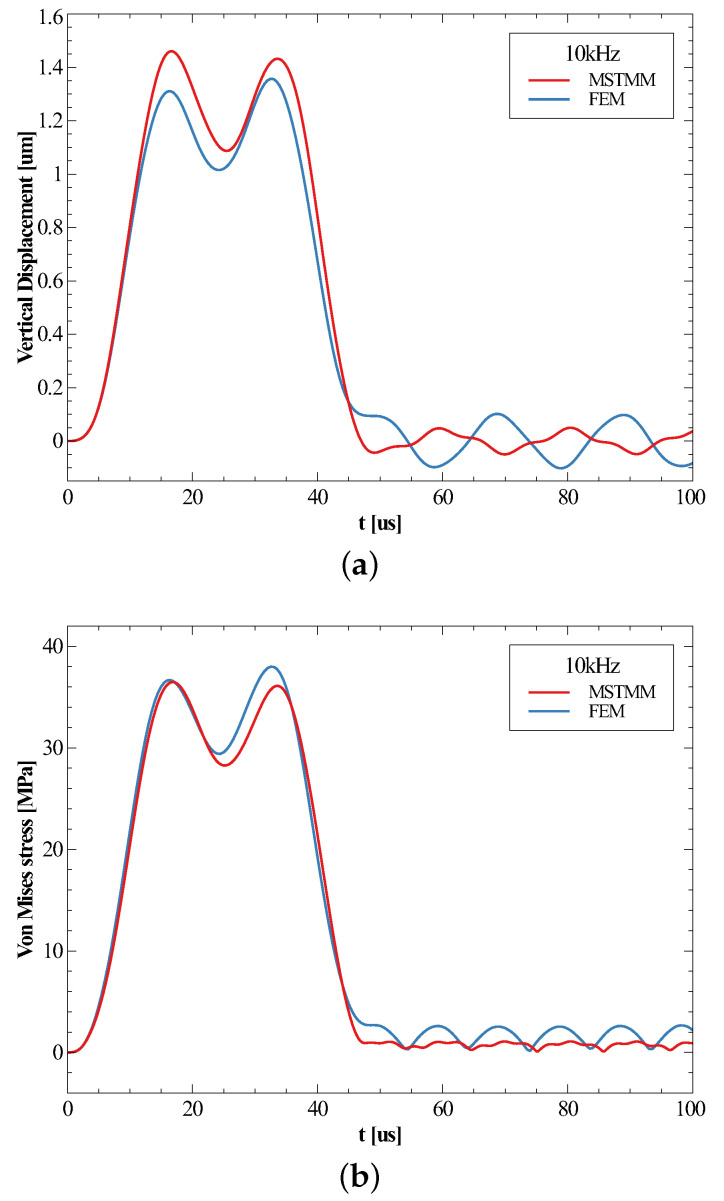
The time history of the maximum vertical displacement (**a**) and the maximum von Mises stress (**b**) under a 10 kHz shock.

**Figure 6 micromachines-14-01187-f006:**
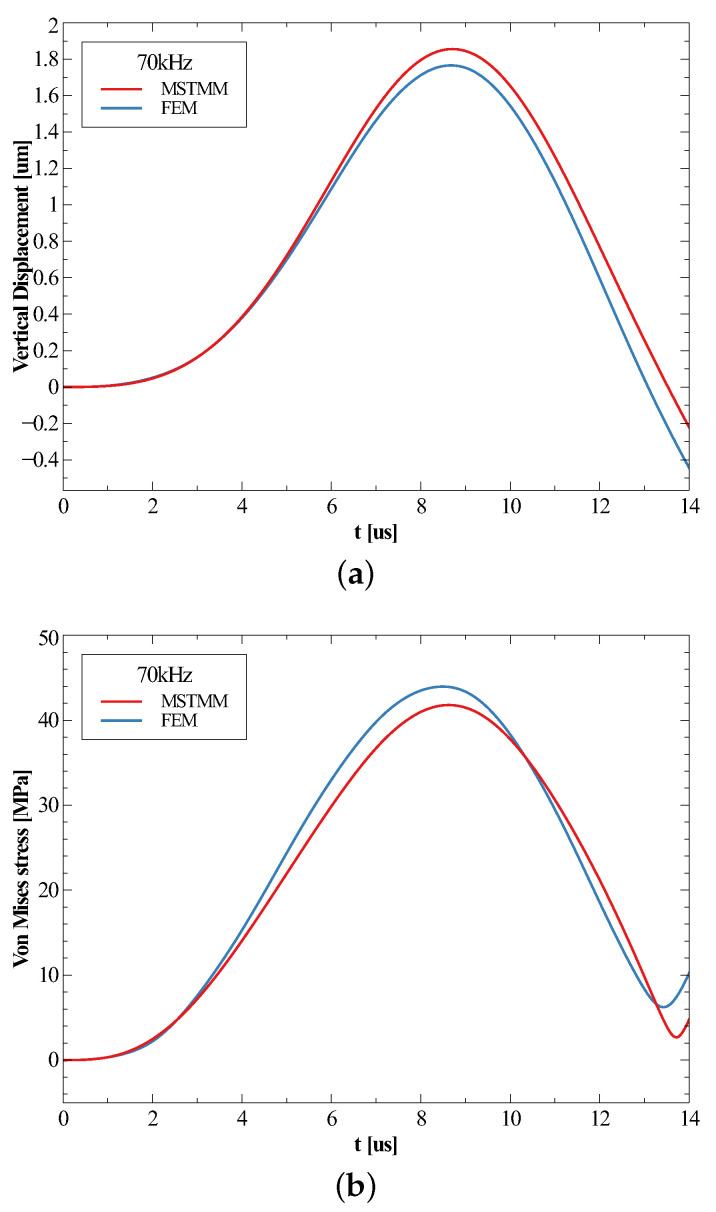
The time history of the maximum vertical displacement (**a**) and the maximum von Mises stress (**b**) under a 70 kHz shock.

**Figure 7 micromachines-14-01187-f007:**
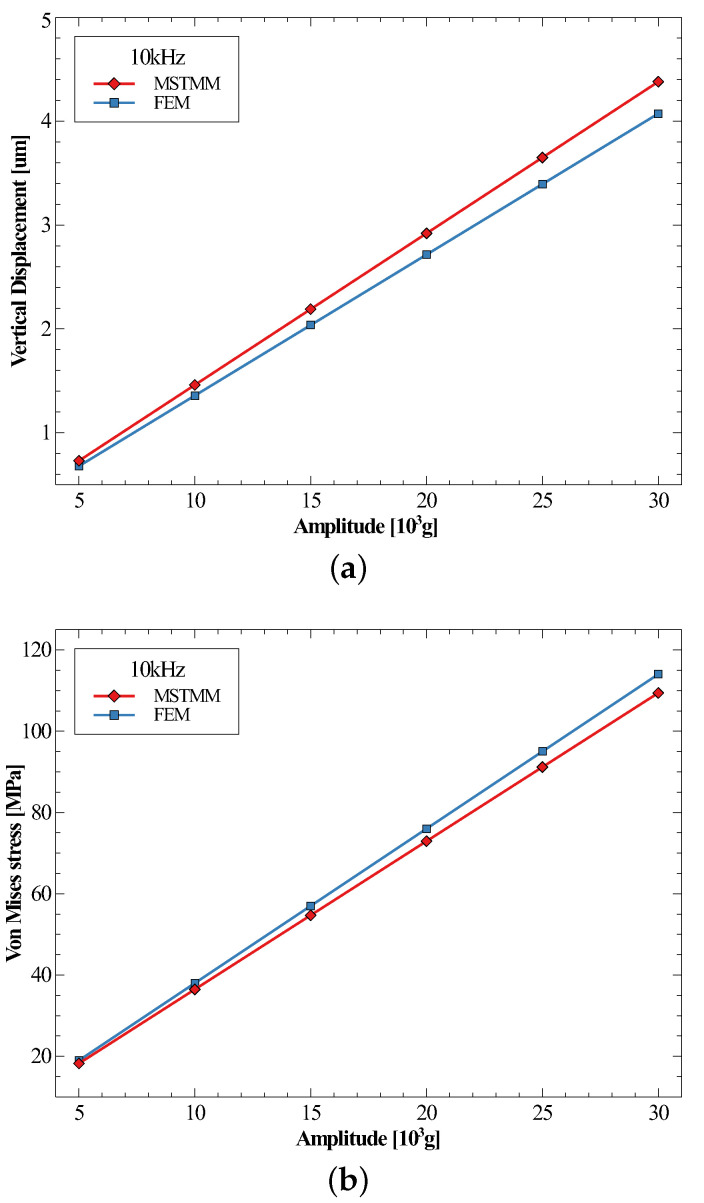
The maximum vertical displacements (**a**) and the maximum von Mises stresses (**b**) under different amplitudes of forces.

**Figure 8 micromachines-14-01187-f008:**
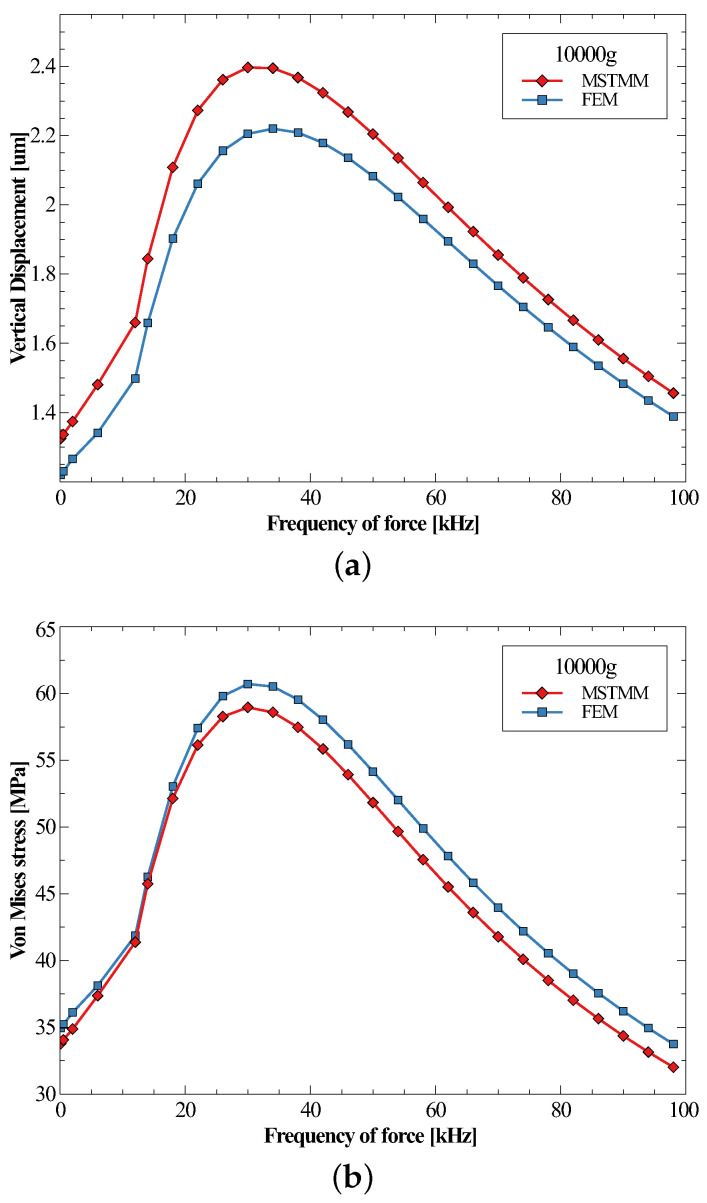
The maximum vertical displacements (**a**) and the maximum von Mises stresses (**b**) under different frequencies of forces.

**Table 1 micromachines-14-01187-t001:** The first three natural frequencies (kHz) obtained by MSTMM and FEM.

Mode	MSTMM	FEM	Relative Deviation
1	49.57	51.01	2.8%
2	72.70	71.85	1.1%
3	139.72	133.19	4.9%

**Table 2 micromachines-14-01187-t002:** The von Mises stress (in MPa) at the five corners of the spiral, obtained using both the MSTMM and FEM.

Corner	MSTMM	FEM	Relative Deviation
P1	26.7	26.3	4.0%
P2	10.6	12.6	11.8%
P3	14.0	17.7	20.8%
P4	27.4	31.0	15.9%
P5	36.5	38.0	1.7%

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
