# Peer review of "Mechanical Response of MEMS Suspended Inductors under Shock Using the Transfer Matrix Method"

_micromachines, 2023, doi:10.3390/mi14061187_

Round 1

Reviewer 1 Report

In this manuscript by Tianxiang Zheng et al., the authors studied the mechanical response of an inductor to a shock load by using the transfer matrix methods of the linear multi-body system (MSTMM) method. The author gave a comprehensive introduction of his model and compared their result with the conventional finite element method (FEM). In a simplified inductor structure, the author’s data suggested the MSTMM methods can show similar results to FEM methods. Overall, I believe this is a comprehensive study, and I recommend publishing this paper.

My minor suggestion is the author should elaborate more on the efficiency of the MSTMM model compared with FEM and why MSTMM can have better efficiency than FEM. And The author maybe should also discuss the possibility of extending this model to a more complex structure. Is this model still valid for more complex structures, and what’s the limitation of this method?

Reviewer 2 Report

This manuscript provides a detailed description of using MSTMM to analyze the dynamic response of a suspended inductor under a high shock acceleration. The system's free vibration characteristics, as well as the dynamic response under shock load, are obtained. This research contributes to the study of the mechanical behaviors of MEMS inductors. Although the subject matter fits well within the scope of the journal and the data provided is extensive, plus the topic is original, there are still several areas of concern that require attention.

1 English should be more authentic, please let a native speaker polish it.

2 What are the advantages of MSTMM compared to the previously reported methods, such as FEM?

3 The manuscript proposes the analysis of the mechanical response of the inductor to a shock load using MSTMM.  Therefore, what is the scientific problem that the study addresses? What is its significance for practical application? Thus, the author should provide a more in-depth analysis of its potential scientific value and practical significance in the last paragraph of the introduction and the conclusion section.

4 The references cited are not up-to-date enough, please include more literature from the past five years.

English should be more authentic, please let a native speaker polish it.

Round 2

Reviewer 2 Report

The author has addressed my questions one by one and made corresponding improvements.  I suggest that the current manuscript deserves publication.